# Soil conditioner mixtures as an agricultural management alternative to mitigate drought impacts: a proof-of-concept.

Juan F. Dueñas[1,2], Edda Kunze[1,2], Huiying Li[1,2], Matthias C. Rillig[1,2]

[1]Institute of Biology, Freie Universität Berlin, 14195 Berlin, Germany
[2]Berlin-Brandenburg center of Advanced Biodiversity Research, 14195 Berlin, Germany

*Correspondence to*: Juan F. Dueñas (juan.duenas@fu-berlin.de)

**Abstract.** Agricultural activities in the northeastern German plains are at risk due to climate change. Soil amendment with conditioners that improve water storage is a plausible alternative to mitigate such risks. While single conditioner additions to soil can be positive, doubts regarding their individual scalability have been raised. An unexplored alternative is to apply
multiple conditioners as mixtures, thus reducing individual application rates while harnessing possible complementarities derived from the interaction of diverse conditioner types. As an initial test of this concept, we implemented a microcosm experiment in which soils amended with none, single and multiple conditioner mixtures were incubated in simulated dry conditions, at constant temperature, for three weeks. We found the addition of conditioner blends consistently and significantly increased maximum water holding capacity and aggregate stability of soils, while decreasing the bacterial to fungal ratio in
relation to soils amended with only single, or no conditioners. These results highlight the practical potential of this concept, and offer the opportunity to develop a robust drought mitigation strategy. We encourage further testing and development of the concept via the implementation of greenhouse and field trials, and the long-term monitoring of the effects using a wider variety of indicators.

## Introduction

Weather extremes (e.g. droughts or heat-waves) could become more frequent in the northeastern part of Germany in the near future (Samaniego et al., 2018). In addition to climate change, shifts in the economic activities in the region, such as the cessation of coal mining planned for 2035, imply a drastic reduction in the amount of water that will be available within the Spree river and associated catchments. This exposes agricultural activities in this region to a significant risk, given a crucial factor determining yield is the access of crops to water. Recent statistics show that in the state of Brandenburg, only 2% of the
area dedicated to agriculture is currently irrigated (Troege and Schulz, 2018). While an increase in the irrigation surface is one possible solution to mitigate drought risks, increased allocation of limited water resources to irrigation is clearly at odds with other economic activities and the safe operating space of natural ecosystems (Daliakopoulos et al., 2016; Wang-Erlandsson et al., 2022). In this scenario, research on access to economically sustainable management alternatives become crucial to mitigate drought-imposed risks.

Agronomy is rich with studies that highlight the positive effects of conditioner addition to water retention, soil structure, and fertility (Diacono and Montemurro, 2010; Edeh et al., 2020; Qin et al., 2015). Popular organic conditioners such as biochar, compost, and straw have been shown to reduce evapotranspiration and improve plant available water (Edeh et al., 2020; Qin et al., 2015; Siedt et al., 2021). Abundant inorganic conditioners, such as silicon-based materials (e.g. amorphous silica), offer an attractive alternative to manage soil texture by increasing the surface area and porosity of coarse textured soils (Schaller et al., 2020; Zarebanadkouki et al., 2022). Biological amendments, defined here as solids or suspensions containing metabolically active microbial communities, can be added to soil to boost crop resistance to drought stress via the establishment of symbioses, or via the improvement of soil structure (Coban et al., 2022). While the adoption of either one of these management options is valid, doubts remain about their individual scalability (Poulton et al., 2018; Rubin et al., 2023; Siedt et al., 2021). That is, the implementation of either one of these conditioners over large surfaces is hampered by their regional availability, production costs and general applicability across different environmental contexts. Even when sourcing individual conditioners might be economically viable, such as the reutilization of farm organic waste or the development of regionally-sourced microbial inoculants, these management practices are not exempt from risks. Examples of risks are the unintended stimulation of potent greenhouse gases (Guenet et al., 2021), the co-introduction of pollutants or pathogens (Wahdan et al., 2023), or the failure of the inoculant to establish (Silverstein et al., 2023). It is thus clear that the costs of implementing individual management options should not only be accounted for in terms of the financial investment required, but also in terms of the environmental costs and risks imposed by the production and distribution of such management options (Rubin et al., 2023; Siedt et al., 2021).

A thus far unexplored alternative to overcome some of these issues is the application of conditioner mixtures in a gradient of increasing complexity. That is, conditioner combinations that consist of a minimum of two conditioner types but increase in complexity with the number of conditioners present in the mixture. A great barrier in the adoption of such management options is the lack of knowledge on the effects of complex conditioner combinations on soil. In contrast to the abundant literature on the effects of single conditioner application, studies rarely test the effects adding more than two conditioner types. Aside from not solving the issue of scalability, this means that scientist have explored less than 1 % of the management options available to increase yield in a sustainable manner (Rillig and Lehmann, 2019). Hence one of our primary goals in this study was to assess whether the addition of conditioner mixtures consistently results in interactions or complementarities that justify its validity, and add value to the more established practice of adding single conditioners to soil. To provide an initial proof-of-concept, we implemented a microcosm experiment where we tested the effect of single and multiple conditioner addition on the water retention capacity of soils, stability of soil structure against water erosion, changes in soil pH and in the abundance of microbes (i.e. bacteria and fungi). In describing our experimental approach, we introduce the reader to a workaround to the combinatorial explosion problem that inevitably arises when dealing with factorial experiments that include more than two factors (Orr et al., 2020; Rillig et al., 2019). We will then discuss the merits of this proposed approach, comparing it to non-amended soils and single amendment alternatives. We discuss the types of interactions that can arise from such mixtures in light of our results and highlight possible interactions or complementarities that arise from adding conditioner blends. We then

call for other researchers to test this concept, measuring a broader set of variables and in more complex systems. Finally, we
briefly propose ways to improve the randomized selection procedure.

**Materials and Methods (Maximum 3 figures and or tables)**

A factorial microcosm experiment, where single and multiple conditioner types were added to soil, was performed for a period
of three weeks between June and July 2023. As microbial populations, and the soil processes these govern, shift in a matter of
hours; the chosen experimental period was deemed suitable to assess treatment effects. The soil employed was collected in a
private field at Briesen (Mark) municipality (52.3860033, 14.2617774). Upon collection, soil was air dried and sieved (2 mm)
to eliminate coarse organic and mineral particles. After sieving, soil was dried at constant 25°C for a period of four days. This
soil is characterized by its coarse texture and can be classified as a sandy loam (Cambic Podzol, Federal Institute of
Geosciences and Natural Resources, 2007). Further soil chemical characteristics are presented in Table 1.

**Table 1.** Summary of experimental soil baseline chemical characteristics

| Parameter[1] | Value | Qualitative reference[2] |
|---|---|---|
| Organic matter | 1 % | Within range for arable soils |
| pH | 6.4 | Slightly alkaline |
| C/N ratio (mass) | 11:1 | Within range for arable soils |
| Phosphorus | 5 mg 100 g$^{-1}$ soil | Low |
| Calcium | 9.2 mg 100 g$^{-1}$ soil | Optimal |
| Magnesium | 7.1 mg 100 g$^{-1}$ soil | High |

[1] Analyses conducted by Agrolab Agrar GmbH according to the VDLUFA standards or locally at our lab with an Elemental Analyzer
(Hekatech, Germany)

[2] Qualitative references are based on the guidelines for soil fertilization of the state of Brandenburg or from (Weil and Brady, 2017)

A panel of five conditioners commonly applied to soil to improve water related properties was selected (Table 1). These
conditioners were selected because they are non-toxic, easily sourced, and have been shown to increase water retention when
added as individual conditioners (see introduction). These conditioners were mixed, or inoculated, in 35 g of sieved soil as
single amendments (n = 8 x 5 = 40), or as three- (n = 10) and five-way (n =10) combinations. Non-amended soil served as
control (n = 10). The constituents of three-conditioner mixtures were randomly selected with replacement from the pool (Table
1) in an iterative process inspired in classical diversity-productivity relation experiments (Tilman et al., 1996). This approach
is a practical workaround to the combinatorial explosion problem (Rillig et al., 2019), which in this case expresses as a steep
growth in the number of replicates needed when ten replicates of all possible three-way combinations are produced (C(5, 3) =
10, n = 100). Furthermore, since single conditioner additions have been thoroughly tested in the literature, and to keep the
number of experimental units manageable, it was decided that each single conditioner treatment would be replicated eight

times only. With these modifications, we were able to run an experiment consisting of 80 units (instead of 180 possible units), without losing statistical power.

Given the premises of this experiment were to reduce the input of any given amendment type and create complex conditioner mixtures, priority was given to find the lowest possible application rate. As application rates of different organic amendments differ widely (Siedt et al., 2021), these were normalized here to 1.5 % (w w$^{-1}$) to facilitate experimental setup and cross type comparison. This compromise represents a rather low dosage for biochar and compost application, and a rather high dose for straw (Edeh et al., 2020; Qin et al., 2015; Siedt et al., 2021). Amorphous silica dose was set to 1 % (w w$^{-1}$) given this is the minimum rate of application that produced a noticeable increase in water retention in prior experiments targeting similar soils (Zarebanadkouki et al., 2022). Microbial inoculum was produced in our laboratory by the dilution method and its application rate was set to 2.7 % (v w$^{-1}$), which is low relative to the rates applied in ecological experiments (e.g. Yang et al., 2021).

**Table 2**. Panel of amendments, in alphabetic order. Unless otherwise stated, all application ratios are expressed in a w w$^{-1}$ basis.

| Conditioner | Type | Provider | Application rate | Chemistry/Biology[1] | Origin |
|---|---|---|---|---|---|
| Biochar LD | Organic | Fetzer GmbH, Eislingen, Germany | 1.5 % | Total C 88 %; org. C 87.6 %; total N 0.56 %; pH (CaCl$_2$) 9.5 | Wood chips, pyrolysis at 550° C |
| Compost | Organic & Biological | Botanical Garden, Berlin, Germany | 1.5 % | Total C 11.5 %; total N 0.68 %; pH (CaCl$_2$) 7.75 | Feedstock consisted of plant residues, biochar (6.7–8.9 vv$^{-1}$), milled rock and clay. |
| Microbial wash | Biological | Gut & Bösel GmbH, Briesen (Mark), Germany | 2.7 % (v w$^{-1}$) | Obtained by filtering (0.53 mm mesh) fresh compost suspension.<br><br>DNA content: 0.42–0.52 ng µl$^{-1}$<br><br>Bacterial to fungal ratio: 1.52 | 200 g of compost in 400 mL of deionized H$_2$O. feedstock consisted of plant and fermented food residues; powdered clay and basalt |
| Amorphous or fumed Silica | Inorganic | Evonik GmbH, Wesseling, Germany | 1 % | SiO$_2$ 99.8 %; pH 3.7–4.5 | Combustion of Si precursors in a hydrogen flame |

| Straw | Organic & Biological | Gut & Bösel GmbH, Briesen (Mark), Germany | 1.5 % | Total C 47.2 %; total N 0.34 % | Rye straw produced post-harvest |

[1] Values presented in this table were in most cases reported by the provider. Molecular and elemental analysis of microbial wash and compost were carried out in our laboratory. Analytical procedures are described in the supplementary material file.

The rate of application of each single conditioner in the three- and five- way mixtures remained constant with respect to the individual rate of application (Table 2). Hence, mixtures had an overall higher application rate than any single conditioner. To disentangle the effects elicited by higher application rates from those elicited by conditioner richness, we included an additional control (biochar, n = 10) in which soil was amended with biochar at an application rate three times higher than the single addition. We selected biochar for this second control as it is mostly consisting of stable carbon (C) polymers, thus avoiding the fertilization effect that will occur when adding amendments containing less stable C, or other nutrients.

Soil amended with the different conditioners were added to 50 mL centrifuge tubes (Thermo Fisher Scientific, USA). To assess the effects of conditioner addition under dry conditions, we incubated our experimental units in a moisture content that corresponds to 30 % of the maximum water holding capacity (mWHC) of control soils. We determined this quantity by the drainage method (Govindasamy et al., 2023). Briefly, eight small plastic containers, each containing 10 g of dry soil, were passively saturated with de-ionized water. Each container was fitted with a fine mesh at its bottom to prevent soil loss. Containers were subsequently placed within a funnel and their upper ends were covered with a moisturized filter paper to minimize evaporative loss. Water drained into a measuring glass for a 24 hours period; then the container was weighted again. The mean mWHC of non-amended soils determined by this procedure was 37.43 % (g g$^{-1}$ of soil). Taking this value as a reference, water was then added to each experimental unit such that all system reached 60 % mWHC, thus allowing microbial communities to become active. To allow a gradual transition from wet to dry conditions, the lids of all units were removed to permit evaporation until they reached the 30% WHC mark (11.2 % g g$^{-1}$, 5–15 days). From that point onward, all units were sealed with a plastic lid, and water content was held constant for three weeks. Water content maintenance was monitored every two days, taking the opportunity to open each unit's lids to allow air exchange. Experimental units were kept in an incubator at constant 20° C until harvest.

At the end of the incubation period, soil was harvested and dried at 25° C for four days. A 4 g aliquot was used to determine aggregate stability against water (Kemper and Rosenau, 1986) by means of a wet sieving apparatus (Royal Eijkelkamp, Netherlands). A 10 g aliquot was used to measure maximum water holding capacity with the drainage method described earlier. A 5 g aliquot was used to measure shifts in soil pH with the CaCl$_2$ method, and 1.5 g of fresh soil were frozen at -20° C to capture the response of soil microbial communities to the experimental manipulations. The later samples were then employed to measure the molecular abundance of bacteria and fungi. The methods employed to measure bacterial to fungal copy per

DNA ng ratio, hereafter refereed as B:F ratio, are described in detail in the Supplementary information document (Materials and Methods).

    All statistical analyses were performed in R 4.1 (R Core Team, 2022). One-way beta (Cribari-Neto and Zeileis, 2010), or linear regressions were employed to assess the effects of adding amendment mixtures on the response variables. The explanatory variable in all models was encoded as "No. of factors", with four different levels representing the number of amendments in

the mix sorted by complexity (i.e. 0, 1, 3, 5). Level 0 here represents non-amended soils, while level 1 represents all the amendment types added individually, irrespective of their identity or dosage. Finally, to assess the effects of individual amendment dosage versus the complexity in the mixture, we specified a second set of models. That is, the first set of models compared amendment mixtures against non-amended soils. The second set compared amendment mixtures to the soil amended with triple dosage of biochar.

Model estimates were then used to estimate marginal means by the least squared method as implemented in package *emmeans* (Lenth et al., 2021). Post-hoc tests of mean differences were performed with Dunnett corrections to minimize type I error. To support the inference of the effects elicited by increasing mixture complexity, mean estimates of each model were plotted in the original variable scale. Uncertainty around these predictions (i.e. confidence intervals at 95%, CIs) were calculated after each model was refitted with bootstrapped data generated with ordinary parametric bootstraps as implemented in package *boot*

(n=5000, Davison and Hinkley, 1997).

**Results**

    Adding amendments to soil, either individually or as mixtures, affected the capacity of soil to retain water inconsistently. The addition of various single amendments elicited heterogeneous responses, ranging from negative to positive (Fig. S2, supplementary materials), but generally leading to minor shifts in mWHC (Z=0.41, p=0.93; Fig. 1A). A closer examination of

the effects of each individual amendment shows straw is the only conditioner leading to a significant increase in mWHC in relation to control (Z=4.6, p<0.001; Fig. S2A). By contrast, mWHC increased strongly when mixtures of increasing complexity were added to the soil. The clearest increase in relation to control was observed when mixtures composed of all five conditioner types were added to soil (Z=4.19, p<0.001; Fig.1A). The positive effects in terms of water retention elicited by adding mixtures was not observed by tripling the dose of biochar single application (Z=-0.25, p=0.97, Fig. 1B).

Soil amendment addition also elicited a clear stabilizing effect on macroaggregates (> 250 µm). The addition of different individual amendments at low dosages generally did not affect aggregate stability (Z=0.41, p=0.93; Fig. 1C). A clear exception to this trend was the addition of straw, which strongly increased the stability of aggregates (Z=5.21, <0.001; Fig. S2B). Mirroring the trend observed on water retention, adding mixtures of increasing complexity consistently elicited stronger increases in the stability of aggregates (Fig. 1C). That is, adding mixtures composed of three to five conditioners led to a

significant increase in the stability of micro aggregates (Z=2.5, p=0.036 and Z=5.08, p<0.001, respectively). While increasing

the dosage of biochar addition elicited a slight increment in aggregates stability (Fig. 1D), this effect was not as clear as the one observed when adding conditioner mixtures (Z=1.28, p=0.433).

The addition of individual soil conditioners to the soil generally increased pH (Fig. S2C, supplementary information). The strongest liming agents among the panel were straw, followed by the addition of biochar (Z=4.24, p<0.001 and Z=3.27, P=0.009, respectively). Interestingly, adding conditioners in mixtures of 3 to 5 factors also generally increased pH in relation to control, but less drastically than individual amendments (Z=1.51, p=0.311 and Z=2.06, p=0.112, respectively; Fig. 1E). Among mixtures, the smallest increase in pH recorded was measured in soils amended with 3 factor blends (Fig. 1E). Increasing the dosage of biochar led to a significant liming effect in relation to control (Z=6.53, p<0.001; Fig. 1F), and to the individual dosage of the same conditioner (Fig. S2C).

Microbial communities generally did not respond to the addition of single amendments (Fig. S3). Although silica addition tended to reduce the B:F ratio, only the addition of straw elicited a statistically supported change in microbial abundance (Z=-5.21, p<0.001; Fig. S3). By contrast, microbial groups responded strongly to the addition of mixtures (Fig. 1G), given reductions in the B:F ratio of increasing magnitude were observed after the addition of 3- and 5-way mixtures (Z=-3.219, p=0.006 and Z=-5.12, p<0.001, respectively). Increasing the amount of biochar added led to a moderate reduction in B:F ratio in relation to non-amended soils (Z=-1.99, p=0.134; Fig. 1H).

All model coefficients and standard deviations are presented in the supplementary information document (Table S1–3)

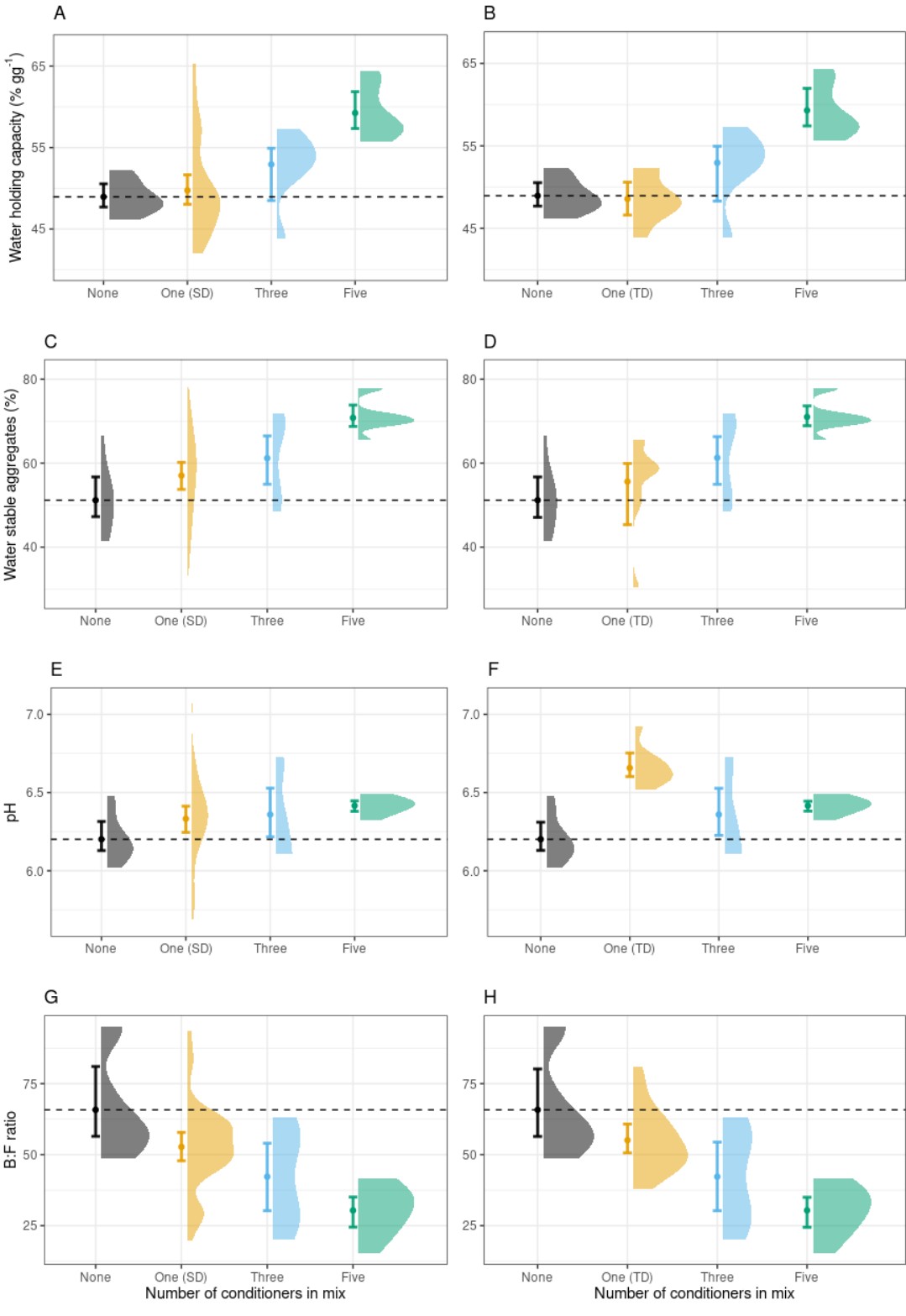

**Figure 1**. Effects of adding conditioners individually or in increasingly complex mixtures on (A) soil water retention, (C) soil aggregate (> 250 µm) stability against water, (E) pH, and (G) B:F ratio in relation to non-amended soils. Conditioner richness was also contrasted against adding a single conditioner (biochar) at three times (TD) the individual dosage (SD) used in any of the mixtures (B, D, F, H). The solid-colored circles represent the predicted means of each treatment level, while the whiskers represent 95% confidence intervals estimated via non-parametric bootstraps (n=5000). The distribution to the right represents a density function of the original data. The dashed horizontal line corresponds to each model's intercept and is plotted for reference.

**Discussion**

Using a microcosm experiment we have shown that multiple amendment addition consistently improves soil's maximum water holding capacity, structural stability and reduces the B:F ratio in relation to soils amended with a single conditioner type. We also showed that while the addition of complex blends moderately increases the pH of soil, the strength of this liming effect is limited in relation to the effect elicited by some individual amendments such as biochar or straw. These results are encouraging as they support the notion that amendment mixtures composed of a wide variety of factors could be considered a plausible agricultural management alternative to mitigate drought-imposed risks in the predominantly sandy soils present in the northern German plains. However, the simplicity of our microcosm experiment does not allow us to generalize these results to more complex systems.

The potential of adding single conditioner types to improve soil water retention and structure, or as liming agents, has long been established (Poulton et al., 2018). Our results are somewhat consistent with these findings, as literature has shown that the magnitude of the effects is sensitive to the application rate (Diacono and Montemurro, 2010; Edeh et al., 2020; Joseph et al., 2021; Zarebanadkouki et al., 2022). Considering a soil with a bulk density of 1.4 g cm$^{-3}$, and a depth of 0.1 m, would mean we applied organic, inorganic, and biological conditioners at a rate of 21 Mg ha$^{-1}$, 14 Mg ha$^{-1}$ and 37.8 m$^3$ ha$^{-1}$, respectively. Most of these rates are effectively among the lower end of application reported in the literature (Siedt et al., 2021; Yang et al., 2021). Given the low application rates, it is not surprising that in many individual addition instances we did not observe significant changes in relation to control soils (Fig. S2). A perfect illustration of this logic are the results observed after the application of straw, which is normally applied at a third of the rate used in this study (Qin et al., 2015; Siedt et al., 2021), yet is the only amendment in our pool consistently showing clear positive outcomes. Though it is possible to attribute the absence of targeted effects of most individual conditioners to their low application rate, tripling the dosage of biochar did not lead to a clear improvement in the soil processes analyzed. In fact, tripling the dosage of biochar and straw led to a drastic pH increase (Fig. 1E). A strong upward pH shift increases the risk of soil alkalization, especially if irrigating with certain types of water (Daliakopoulos et al., 2016; Weil and Brady, 2017). Thus, rather than exclusively representing the magnitude of dosage, our results signal that the effects elicited by amendment mixtures result from interactions between conditioners present in those mixtures.

We base our discussion of the possible interaction types suggested by our data on the knowledge about the mode of action of each individual amendment on the variable of interest. We consider four main classical interaction types outlined in the multiple factor literature (Orr et al., 2020). That is, we consider synergism, antagonisms, additivity and dominance as possible joint factor outcomes. With this in mind, at first glance the effects elicited by conditioner blends on water retention, aggregate stability and B:F ratio, would appear to be driven by a dominative effect of the best performing individual amendment: straw.

The positive effects of straw on soil water retention are often explained by a stabilization of soil temperature which in turn leads to a reduction in evapotranspiration (Qin et al., 2015). Given we kept temperature and moisture levels constant during our experiment, and straw was not applied as a mulch layer, this explanation would not fit the outcomes we observed. Instead, an antagonism between organic matter and soil microbial diversity enhancement appears to better characterize the outcomes. There is ample evidence indicating that higher soil biodiversity positively influences many soil process rates (Wagg et al.,

2014; Yang et al., 2021). It is then reasonable to think that when biological inoculants enhance the biodiversity of an unfertile soil, such microbial populations would be obligated to adopt dormancy, or use the limited resources available to them. A reduction in microbial activity, and even further depletion of organic matter would in turn lead to a reduction the structural complexity and sorption capacity of soil, which ultimately hinders its capacity to retain water (Fig. S2A). By contrast, when organic matter in combination with diversity enhancers are added, microbial activity would increase, leading to an increase in

soil water retention through the combined action of physical and biological mechanisms. Similar to the argument made for water retention, a strong increase in the stability of soil aggregates in the soils amended with mixtures would be better explained by an antagonistic interaction between organic matter and bioinoculant addition, rather than by a dominative effect of straw. While there are studies that certainly show that straw addition alone leads to an increase in the stability of aggregates (e.g. Rahman et al., 2018), this outcome is often explained as a byproduct of the stimulation of microbial activity by the addition of

organic matter to soil. In absolute terms, it is clear that a greater resource availability might have resulted from the concurrent addition of biochar, compost, straw, than by the sole addition of straw (Table 1). Furthermore, we observed a strong shift in B:F ratio, particularly driven by an increase in fungal abundance, when diverse carbon sources were added together with biological enhancers (Fig. 1G). An increase in fungal abundance in relation to bacteria is congruent with the addition of a microbial inoculant characterized by a low B:F ratio (Table 1), a buildup of organic matter in low fertility soils, a greater

degree of soil aggregation, and a stimulation of straw associated saprobe communities (Wahdan et al., 2023). Though the mean increases in water holding capacity, aggregate stability and fungal abundance observed when adding mixtures were not significantly greater than the increase observed when adding straw alone, adding multiple nutrient sources and microbial inoculum certainly contributed to reducing the variability of the outcome (Fig.1, Fig. S2). Finally, antagonistic interactions of a different kind also appear to fit the outcomes observed in terms of pH shifts. While biochar, compost and straw tend to be

alkaline; microbial inoculants and silica are characterized by their neutrality or acidity, respectively (Table 1, Fig. S2). Thus, organic amendment tendency to make the medium alkaline might have been neutralized to a certain extent by the presence of the silicon dioxide. Interestingly, partial support for the discussed antagonisms can be visualized in the positioning of the three-way mixtures (Fig. S3). Although this exercise does not intend to replace a formal test, it can be seen that the samples amended

with mixtures that scored the highest mWHC and WSA values, or the lowest B:F ratios, tended to contain two organic amendment types and one unit of inoculum. Likewise, in terms of pH, the samples that did not contain $SiO_2$ scored the highest. Altogether, this evidence suggests that antagonisms between the conditioner in the mixtures drove and stabilized the positive changes observed, while at the same time dampening potential negative effects in terms of alkalinity.

Here we focused on specific soil indicators that signal an increase in the resistance of soil to drought stress. Yet, a more balanced assessment of the merits of this approach must necessarily look at a broader set of response functions. Indicators that assess possible undesired side effects, such as increased nitrous oxide emissions, uncontrolled increases in sodicity or pH, introduction of contaminants or pathogenic agents and yield reductions must be included in further experiments (Guenet et al., 2021; Joseph et al., 2021; Rubin et al., 2023; Wahdan et al., 2023). Future trials shall confirm the observed improvements using more accurate metrics such as retention and hydraulic conductivity curves. The risk of unintentional increases in soil alkalinity, or sodicity, should be carefully considered. Sodicity might arise when soil amending increases the retention of irrigated water, which is then rapidly lost by evaporation, and ultimately may lead to the accumulation of salts in the rooting zone. Therefore, we recommend any future trial involving multiple amendment addition in combination with irrigation must closely assess electric conductivity, cation exchange capacity or the sodium adsorption ratio (Weil and Brady, 2017). Notwithstanding these reservations, our results indicate that one of the attractive features of adding conditioner mixtures is that they not only ameliorate the targeted function, but show complementarity effects. With complementarity, we mean that aside from a clear effect on increasing water retention, which is arguably the most direct indicator of drought resistance employed in this study, mixture addition elicited positive effects in soil structure and microbial communities while at the same time appeared to buffer soils against drastic shifts in pH. As discussed above, further complementary functions of adding mixtures could arise from the buildup of organic matter in soil and its interactions with microbial diversity. For instance, another complementary outcome highly relevant to the quality of water in the region is that adding mixtures instead of single amendments might result in greater reductions in the leaching of nutrients or polluting agents to freshwater (Joseph et al., 2021; Qin et al., 2015; Siedt et al., 2021). To assess if this is the case, similar controlled trials in greenhouse settings are necessary as these could reveal a broader set of complementarities, or tradeoffs (Guenet et al., 2021; Poulton et al., 2018; Wahdan et al., 2023) resulting of applying conditioner mixtures.

Other important aspects in need of validation are the economic scalability of this practice, and temporal variations in the manifestation of effects. Scalability is a crucial aspect to tackle, given it paves the way for the development of management practices with the potential to reach a much wider number of actors. On the other hand, effect variation over time is an important logistical factor to assess, as it informs the frequency with which amendment mixtures need to be re supplied to soil to stabilize the effects. We envision temporal variations to occur, given some of the conditioners in our mixtures are, by nature, more durable in soil than others (Diacono and Montemurro, 2010; Joseph et al., 2021; Rahman et al., 2018; Schaller et al., 2020). For instance, biochar or silicon persist in soil for a significantly longer period of time than straw (Wahdan et al., 2023), or microbial inoculum (Silverstein et al., 2023). While it is well known that biochar can deliver positive effects over long time periods (Lehmann et al., 2003), it is a variable material, and there is still no solid understanding of the long-term effects of its

repeated application (Joseph et al., 2021). In contrast to biochar, research on synthetic amorphous silica application in agricultural settings is recent. Given there is a natural pool of biogenic available amorphous silica in soils, and it is also accumulated in plant tissues (Schaller et al., 2020), future supplementation of synthetic silica mixed with plant residues must account for these background levels. The challenge is then to monitor the effects elicited by amendment mixtures over longer time periods and to accommodate the formulation by considering the residence time of its components and the background availabilities of these materials in plant residues and soil. A conservative strategy could be to design mixtures composed of a bigger proportion of labile elements in relation to more stable ones, thus minimizing the uncontrolled buildup of more durable materials in soil.

We have already recognized the limitation of factorial experiments when it comes to dealing with the effects of multiple factors. By necessity we adopted an approach to select elements in our three-way mixtures that has been used elsewhere with success (Rillig et al., 2019). An obvious disadvantage of such an approach is that it opens the possibility that not all possible mixture combinations are present in the final selection at a higher degree of replication (Fig. S3). Moreover, one might end up stochastically selecting a greater number of combinations that contain one amendment to the detriment of another. Such effects could be countered by slightly modifying the re-sampling procedure in such way that it penalizes the frequent selection of one factor. For instance, by establishing a cap on the number of times a particular factor is represented in a mixture, after which the factor is removed from the selection pool. Alternatively, co-design practices (*sensu* Rillig et al., 2024) can be implemented. During co-design, local interested parties together with experts characterize each individual conditioner in multiple dimensions (e.g. cost of application, magnitude of individual effect, possible antagonisms). These multiple dimensions are subsequently collapsed into a score that is finally used to weight each individual factor in the mix. Despite the obvious caveats that random selection brings, re-sampling of factors might bring some unforeseen benefits. One is that this approach tacitly acknowledges the fact that a collective of individuals as heterogeneous as the farming community in the region is very unlikely to adopt a standardized mixture to apply to their fields.

**Conclusion**

The addition of multiple amendment mixtures shows potential as an agricultural management tool to counter the risk imposed by drought, particularly for sandy soils present in the northern German plains. The addition of amendment mixtures representing a higher richness of amendments clearly ameliorated the capacity of soil to retain water under dry conditions, increased aggregate stability and fungal abundance, while holding a similar bacterial abundance and pH relative to non-amended soils. Surprisingly, the addition of complex amendment mixtures did not result in a much greater effect size relative to the single addition of straw, yet mixtures produced less variable outcomes and exhibited a wider range of complementarity in terms of soil functions affected. Additional work is necessary to further confirm the potential of this management option. The economic viability of the practice must be critically assessed and compared against current alternatives. Multiple amendment mixtures should be tested in more realistic agronomic conditions, over longer time periods and including a wider

variety of indicators. We propose that limitations of the experimental design used here can be overcome to a certain extent by a more participatory and deliberate panel selection procedure.

## Code and data availability

The R code employed for statistical analysis and plotting figures as well as all the data produced in this experiment are available under the DOI: 10.5281/zenodo.13221507.

## Author contribution

JFD designed the study, assisted in the experimental setup and data collection, wrote the code to analyse data and the full draft of the manuscript. EK designed the experiment, conducted it, collected samples, analysed soil and reviewed the manuscript. HL assisted with the experimental design, setup, data collection, soil analysis and reviewed the manuscript. MCR obtained financial support, sanctioned the experimental design and revised the manuscript.

## Competing interest

The authors declare that they have no conflict of interest.

## Acknowledgements

This work was possible thanks to the collaboration of Max Küsters, Laurenz von Glahn, Robert Wagner, Helga Kanda and Rebecca Rongstock, who provided materials and feedback for the design of the study. JD would like to acknowledge Asım Can Albay for laboratory assistance, and Daniel Lammel for assistance with protocol development. Two anonymous reviewers provided valuable comments and suggestions to improve the quality of this manuscript.

## Financial Support

Funding for this work was received from the Einstein Foundation Berlin and the Berlin University Alliance (ERU-2020-609).

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
