# Peer review of "Soil conditioner mixtures as an agricultural management alternative to mitigate drought impacts: a proof-of-concept."

_EGUsphere, 2024_

## Author Response (AR1)

Berlin, December 2nd 2024

Dear Editor,

We hereby submit an updated version of the detailed point-by-point reply to both reviewer's comments. Please note that in addition to the changes indicated in the replies, we have amended the title of the manuscript, the acknowledgements section, and we have included four more references. All these changes can be located easily following the reference in each individual answer and using the marked-up version of the new manuscript.

Best Regards,

Juan F. Dueñas, PhD.

**Point by point reply to reviewers' comments**

|  | **Reviewer 1** |
| --- | --- |
| Comment | The article " Soil conditioner mixtures as an agricultural management alternative to mitigate drought impacts" is well presented and addresses a subject of environmental relevance and interest to the scientific community. The experimental procedures adopted are aligned with the precepts of the scientific method. The results are well discussed, and the conclusions are coherent. |
| Answer | We would like to thank the reviewer for these positive comments. |
| Comment | **Lines 60-61**: Why was the duration of the experiment three weeks? What is the rationale for this trial period? |
| Answer | Thanks for your question. We designed this experiment to be an initial proof-of-concept. In that spirit we selected a simple microcosm system, which is attractive as it is easier to control and allows us to focus on soil processes and microbial organisms without having to consider the nuances introduced by the |

presence of a plant. Since it is reasonable to think that at the scale of this experiment, changes in soil processes and microbial taxa occur in a matter of hours to days, we considered the three-week trial period as suitable to assess treatment effects, hence showing the potential of our concept. On the other hand, as the experiment seek to simulate a moderate drought period, we considered that a drought that takes place for more than three weeks would not fall within this category, at least for the area of Berlin-Brandenburg. To clarify the former point to the readers we have now included a justification statement at the methods section (P3, L68-69) of the revised manuscript.

| Comment | **Lines 166-173**: The mixture of conditioners promoted what we call "emergent processes or properties" – when several components compete to enhance attributes that were not expressed in the presence of just one of them. What role does each conditioner play in this tangle of beneficial effects on the soil? Of the conditioners evaluated, is it possible to establish the sequence of importance of each of them? |
|---|---|
| Answer | This is an insightful remark and we thank the reviewer for the questions it raises. In our previous research with multiple stressors on soil, we have not been able to conclusively establish what role each of the individual components in a mixture is playing when it comes to the expression of a certain result. We have been able to establish that over a certain number of components, the system appears to be 'steered' in either a positive or negative trend. There are certain statistical methods that allow us to estimate whether the interaction of two factors is additive, multiplicative or antagonistic. However, there are no such methods available for interactions that include numerous factors. The latest research in our lab shows that effect similarity and nature of individual components is a promising predictor of the magnitude and direction of the effect when present together in a mixture. |

| | In summary, more research is needed to establish the role each individual factor plays when present in a mixture. At this point we can only propose which mechanism could be at play, which we believe we did in the discussion section (P10, L210-274). But in the absence of a good method to untangle the contribution of individual amendments, we prefer to use a more tentative language and leave the discussion as is. |
|---|---|
| | **Lines 247-248**: "For instance, biochar or silicon persist in soil for a significantly longer period of time than straw (Wahdan et al., 2023)". Given the possibility of longer residence time in the soil, researchers consider the possibility of obstruction of soil pores by both materials. If this risk exists, the authors can suggest how to manage the application of conditioners? |
| Answer | Thanks for another insightful comment and question. As far as we are aware, biochar can be incorporated into soil aggregates and is a porous material. Therefore, we do not think that pore obstruction is a major risk in that particular case. Having said this, there are still major gaps in our knowledge about the long-term effects of repeated application of biochar in soils. Amorphous silica is also a porous material, has a large surface area, and biogenic pools of this material can be naturally found in soils. So, we think the risk of pore obstruction is low. However, we believe that a long-term investigation of the supplementation of synthetic amorphous silica in combination with plant residues in a soil would be desirable. To highlight these uncertainties to the reader, we have added new text in the discussion section (P11, L276-285). |
| | **Reviewer 2** |
| Comment | The article is written in an understandable language an it is very interesting, I have almost no comments regarding the editing. |

| | |
|---|---|
| Answer | We thank the reviewer for the positive comments. |
| Comment | But:

a) The authors only evaluated water retention capacity. For a stronger statement, other parameters should have been presented, such as soil porosity, hydraulic conductivity, among others. It is weak point of it; |
| Answer | We thank the reviewer for this observation, with which we agree. We deliberately chose a fast, yet least accurate method to determine water retention capacity because the scope of the paper was to create a proof-of-concept trial and given the amount of soil in our experimental units was rather small (35 g each). We generally agree that the parameters that the reviewer is suggesting can serve to build a much stronger case to support the potential of our proposed drought adaptation strategy. However, as the reviewer might be well aware, to obtain such parameters would demand greater amounts of soil which we did not envision for this particular experiment.

In summary, we agree that our approach is limited in the level of generalization it allows. Yet we believe it serves to illustrate the practical potential of our idea. To emphasize these points even further to the reader, we have 1) edited the title of the manuscript which now reads: "Soil conditioner mixtures as an agricultural management alternative: a proof-of-concept"; 2) we have edited a statement in the introduction that clarifies better the scope of the experiment (P2, L54-57); and finally, in addition to the lines at the introduction (P2, L59-61) and discussion (P9, L192-193) that speak about the limitations of this approach, we have included the suggested metrics as possible parameters to measure in future experiments at the discussion section (P10, L250-257). |

| | |
|---|---|
| Comment | b) In my opinion, the methods should be described in more details, as well as the choice of soil conditioners and the treatments (amendment mixtures) evaluated; |
| Answer | Regarding the amendment characteristics, we believe we provide plenty of information in both the introduction (P2, L31-39) and method (Table 2, P4 L99-102) sections. We further provide a succinct statement of the rationale to select these amendments at the methods section (P3, L79-81). We are unsure if the last part of the reviewer critique refers to how the treatments were analyzed (i.e. statistical procedures) or which treatments were included in the experiment. If the answer is the latter, then we believe we have explained this clearly in the methods section (P6, L132-145) which together with the main figure presented, should clarify to the reader which treatment effects were actually assessed. |
| Comment | c) When using Soil conditioner mixtures, changes in soil electrical conductivity may affect plant growth. This could be further explored; |
| Answer | This is a very insightful comment. We agree with the reviewer that this would be an interesting parameter to measure in an experiment that involves plants and irrigation. As far as we understand, electric conductivity is a proxy to assess soil salinity, which indeed affects plants and most commonly arises due to poor soil irrigation practices. Since none of our amendments contains large amounts of sodium, we do not see that the addition of these particular amendments increases the risk of altering soil electrical conductivity. However, we do identify a risk of alteration of electric conductivity when retention of irrigated water is enhanced. Consequently, we have added a statement in this sense in the discussion section (P11, 250-255). |

| | |
|---|---|
| Comment | d) The economic costs are crucial to the viability of the proposed scheme. Further discussion is recommended; |
| Answer | We thank the reviewer for this suggestion, with which we fully agree. One of the primary motivations for developing this procedure arose because of the doubts in the economic scalability of applying single amendments. However, given the absence of data and to avoid unnecessary speculation, we limit ourselves to discuss this point succinctly in the discussion section (P11, L269-71), and encourage future research in this direction at the conclusion section (P12, L308). Furthermore, we believe that an economic assessment is beyond the scope of this piece. |
| Comment | e) The researchers acknowledge the limitations of the research due to the short observation time and the scale on which the experiment was conducted. However, to conclude on the suitability of the conditioners tested based solely on the analysis of the parameters presented is a major oversimplification. |
| Answer | We thank the reviewer for this comment. We would like to clarify that nowhere in our manuscript do we present the observation time and scale of the procedure as a limitation. We simply recognize that, of the large arsenal of procedures available to an ecologist, microcosm experiments offer some advantages and present some obvious limitations. We apologize if, by reading our manuscript, the reviewer got the impression we are concluding our scheme to be ready for wide scale application. We would like to direct the attention of the reviewer to the new title of the manuscript, and to lines (P2, 56-57) in the introduction, (P9, L192-93) in the discussion and (P10, 248-49) in the conclusion sections of the new version of the manuscript. We believe these lines make abundantly clear the preliminary nature of this concept. In other words, we are convinced that further testing is necessary to fully confirm the merits of the procedure. |

| | |
|---|---|
| Comment | My detailed comments are below:

Abstract: I recommend rewriting the abstract with a brief description of the methodology, results and conclusion. The introduction should be shorter; |
| Answer | Thanks for the suggestions. In this new version we have expanded the abstract as recommended (P1, L8-18). Regarding the introduction, we believe it contains essential details about the amendments and is not long. |
| Comment | The text in lines 65 and 66 could be placed after the description of the soil collection in line 63; |
| Answer | We agree with this suggestion and we have now placed these lines after (P2, L70) in the new manuscript version. |
| Comment | The results of the soil chemistry analysis (lines 62-65) could be presented in a table; |
| Answer | We agree with this suggestion and we have now summarized the soil chemistry analysis on a table (P2, Table 1, L74). |
| Comment | Line 68: Why was the number of repetitions for the treatments of single amendments lower (n=8)? |
| Answer | A concise answer is to keep workload manageable during the experimental upkeep. Since the effects of single amendment addition had been thoroughly tested in the literature, and given the large number of experimental units that would have needed to be included due to: 1) the number of amendments in the panel (5), and 2) amendment combinations (please see P3, L83-87), we found a compromise by reducing the number of replicates in the individual addition instances (from 50 to 40), and by only |

|  | including random combinations of 3-way amendments. In other words, we monitored 80 instead of 180 possible experimental units without loss of statistical power. We have included a statement in the methods section of the new manuscript to justify this choice (P3-4, L86-90). |
| --- | --- |